# How can primary care enhance end-of-life care for liver disease? Qualitative study of general practitioners' perceptions and experiences

Holly Standing,[1] Helen Jarvis,[1] James Orr,[2] Catherine Exley,[3] Mark Hudson,[2,4] Eileen Kaner,[1] Barbara Hanratty[1]

[1]Institute of Health and Society, Newcastle University, Newcastle upon Tyne, UK
[2]Institute of Cellular Medicine, Newcastle University, Newcastle upon Tyne, UK
[3]Faculty of Health and Life Sciences, Northumbria University, Newcastle upon Tyne, UK
[4]Liver Unit, Freeman Hospital, High Heaton, Newcastle upon Tyne, UK

**Correspondence to**
Professor Barbara Hanratty;
barbara.hanratty@newcastle.ac.uk

## ABSTRACT

**Background** Liver disease is the third most common cause of premature death in the UK. The symptoms of terminal liver disease are often difficult to treat, but very few patients see a palliative care specialist and a high proportion die in hospital. Primary care has been identified as a setting where knowledge and awareness of liver disease is poor. Little is known about general practitioners' (GPs) perceptions of their role in managing end-stage liver disease.

**Objective** To explore GPs' experiences and perceptions of how primary care can enhance end-of-life care for patients with liver disease.

**Design** Qualitative interview study, thematic analysis.

**Participants** Purposive sample of 25 GPs from five regions of England.

**Results** GPs expressed a desire to be more closely involved in end-of-life care for patients with liver disease but identified a number of factors that constrained their ability to contribute. These fell into three main areas; those relating directly to the condition, (symptom management and the need to combine a palliative care approach with ongoing medical interventions); issues arising from patients' social circumstances (stigma, social isolation and the social consequences of liver disease) and deficiencies in the organisation and delivery of services. Collaborative working with support from specialist hospital clinicians was regarded as essential, with GPs acknowledging their lack of experience and expertise in this area.

**Conclusions** End-of-life care for patients with liver disease merits attention from both primary and secondary care services. Development of care pathways and equitable access to symptom relief should be a priority.

### Strengths and limitations of this study

► To the best of our knowledge, this is the first study to focus on the experiences of primary care clinicians around managing end-of-life liver disease.
► General practitioners (GPs) were recruited from a range of both rural and urban UK locations.
► Use of semistructured interviews allowed us to collect detailed descriptions of GPs' experiences of managing end-of-life liver disease.
► The study was conducted by a team of experienced researchers with a range of disciplinary backgrounds including clinical experience.
► The findings of our study are based on the reported knowledge and experiences of self-selecting participants, their views may not be transferable to the wider GP population.

## INTRODUCTION

End-stage liver disease is an important challenge for providers of palliative and end-of-life care. Death rates have increased 400% since 1970, and it is now the third most common cause of premature death in the UK.[1–3] In 2012, around 11 000 deaths were attributed to liver disease in the UK. Patients have a complex and heavy symptom burden that is often poorly treated[4 5] and the psychosocial needs of patients and families may be considerable.[6] The majority of patients present late to services, when liver disease is irreversible and around 70% die in hospital.[7] The typical clinical course, of gradual decline punctuated by episodic decompensation, may mean that treatment is focused on prolonging life and a palliative care approach is rarely considered.[8] Transplantation is an option only for selected patients,[9] with some evidence suggesting that patients who are considered and rejected for organ transplantation are unlikely to receive any palliative care.[10] Research suggests that people with liver disease are less likely to be involved in end-of-life discussions and planning than cancer patients, though data are limited.[11] Discussing care plans that acknowledge the proximity of death is difficult, particularly when patients are receiving interventionist care. However, it is important, as timely referral to palliative care can be compromised when the focus remains inappropriately on treatment with curative intent.[12]

There is a growing international consensus that end-of-life care for patients with liver disease requires improvement.[4 11 13 14] Primary care has been identified as a specific area where care could be enhanced, particularly around the discussion of palliative care needs with patients, the inclusion of patients on palliative care registers and improving communication between primary and secondary care.[8 15] Primary care professionals are well placed to provide holistic care that patients dying with liver disease need but are seldom involved. In addition, general practitioner (GP) care for patients dying with liver disease is not rated highly by bereaved relatives.[8 16]

Little is known about how health professionals in primary care see their role in end-stage liver disease or what they view as the main barriers to providing high quality care. This study intends to contribute to our understanding of this patient group and to inform the development of appropriate services. The aim is to explore GPs' experiences and perceptions of end-of-life care for patients with liver disease.

## METHODS
This study employed qualitative methods, involving semi-structured interviews with GPs from five geographical areas within England.

### Participants
A heterogeneous purposive sampling approach was employed to ensure that a variety of perspectives and experiences of management of liver disease were sampled; for example, previous management of an end-of-life liver patient, views on whether management should be primary care or secondary care led, as well as a range of practice sizes and localities. Participants were recruited via National Institute for Health Research Clinical Research Networks (CRN) and local networks of GP practices in London, Thames Valley, Wessex, Yorkshire and the North East of England. Sampling began with one CRN and was expanded during the course of the study to include four additional areas. Coordinators at the CRNs were used to target practices in a variety of rural and urban locations, as well as areas of high and low prevalence of liver disease and substance misuse. Email invitations were sent to GP practices within these networks, and GPs who wished to participate then contacted the research team.

### Data collection
A semi-structured interview guide was developed to cover issues identified through a review of the literature, including GPs' experiences of identifying and managing end-of-life liver disease. This was a 'living' document that evolved throughout data collection to allow exploration of emerging areas. Interviews were conducted face to face (n=2) or over the telephone (n=23). Interviews lasted between 15 and 50 min and were all conducted by the first author between March and August 2016. Field notes were taken to aid subsequent analysis. Informed consent was obtained from all participants. Data collection ceased when no themes were emerging from the interviews (see below for further detail).

### Data analysis
Audio recordings of interviews were transcribed verbatim by an independent transcription company; transcripts were checked for accuracy by listening again to each recording. The NVivo V.10 software package was used to manage the data.

Data collection and analysis ran concurrently throughout the study. Analysis of early transcripts informed the interview schedule for later interviews and each transcript was re-examined in light of subsequent interviews. A thematic analysis was conducted.[17] The first stage involved researchers familiarising themselves with the data through detailed reading of the transcripts followed by line-by-line coding.[18] Field notes taken during data collection were used throughout analysis to enhance the reflective process. Several quality control measures were employed to ensure trustworthiness of the data. A proportion of the transcripts (20%) were coded independently by three researchers, before coming together to compare their analysis. Data analysis and emerging themes were also discussed among the wider research team, which included individuals with clinical expertise in general practice and hepatology.

## FINDINGS
Twenty-five GPs were interviewed. The majority had been qualified as GPs for 5 or more years, but few (4/25) had any specialist hepatology or gastroenterology training or experience. Participant characteristics are shown in table 1.

Four themes were identified from the data analysis: the role of the GP, acknowledging and accepting end of life, collaborative care pathways and social relationships and consequences. The quotations presented below are illustrative, representing typical participant responses and demonstrating the varied viewpoints.

### The role of the GP
In this study, few of the interviewees had extensive first-hand experience of managing patients with liver disease at the end of life. Those who did, reported that they managed such cases infrequently, and some years may go by without them seeing a case.

> (We manage) a lot of dying people, but not from the hepatology point of view. I don't know if they tend to be managed in hospital predominantly more than in primary care? That's a possibility, I guess. (GP 7)

Some of the interviewees attributed their lack of expertise and experience of caring for liver patients at the end of life to a reluctance among hospital clinicians to relinquish control.

**Table 1** Participant characteristics

| Characteristic | Number of GPs |
|---|---|
| Sex | |
| Male | 12 |
| Female | 13 |
| Years of experience as GP | |
| <5 years | 5 |
| 5–10 years | 10 |
| 16–25 years | 9 |
| >25 years | 1 |
| Specialist hepatology/gastroenterology experience or training | |
| Yes | 4 |
| No | 21 |
| Size of practice (registered patient population) | |
| <5000 | 5 |
| 5–10 000 | 9 |
| 10 000–15 000 | 9 |
| >15 000 | 2 |
| Geographical area | |
| North West London | 7 |
| Wessex | 8 |
| North East and North Cumbria | 5 |
| Yorkshire and Humber | 1 |
| Thames Valley and South Midlands | 4 |

GP, general practitioner.

There are some conditions, like liver disease, renal failure, they [patients[ all just end up dying in hospital for some reason. I don't know whether it's the hospital consultants that don't want to let them go home… They need to let go and make sure there's a palliative care plan in place…they don't do it. (GP 3)

The limited contact between GPs and patients dying with liver disease was attributed to an unpredictable disease trajectory with periods of stability and decompensation and to patients remaining under the care of hospital services in their last weeks and months. The GPs in this study shared a view that end-of-life care is a core component of primary care, and interviewees questioned how appropriate it was for specialist hospital clinicians to take a lead in palliative care. Patients with liver disease were not regarded as distinct or different from patients dying with other conditions, and a number of GPs expressed a desire for greater involvement in their end-of-life care. Some participants implied that primary care involvement may support more patients to die at home rather than in hospital.

I think primary care probably is best placed, in most cases, to look after people- well not only for that [liver disease], for most end-of-life care issues. So, yeah, I think the GP is probably the most important person in the sense that they can bear in mind what the specialists have advised, but at the end of the day, try and keep some of these patients at home rather than having to have them admitted acutely. (GP 14)

### Acknowledging and accepting end of life

Judging when a patient with liver disease is nearing the end-of-life was perceived to be a particular challenge. Communication about prognosis and the age of patients were identified as important factors. Some of the GPs reflected on how management decisions taken in hospital send out messages that influence care provided in the community. Continuing to pursue active treatment may convey optimism about the patient's life expectancy. Specifically, GPs referenced occasions where patients had been placed on the waiting list for a liver transplant, which the patient saw as offering them a second chance at life even though they were critically unwell and may die while waiting for an organ. Patients with end-stage liver disease are often younger than the typical palliative care patient.[19] It may be that clinicians are more reluctant to give up on active treatment for younger patients,[20] while patients and families may also struggle to accept that the end-of-life is approaching.[21]

Those patients where it's a, kind of, grey area about whether they're end-of-life or not. And I think that mainly stems from the fact that if it's a young patient, it's more difficult for healthcare professionals, the patients themselves, and families, to actually accept that the person's dying. (GP1)

Mixed or uncertain messages may mean that care is compromised, if no one engages the patient in discussions about the end of life, and a palliative approach is never considered.

I suppose, looking back it really was palliative care but they (secondary care) put him on the transplant list because he's given up alcohol and there was still this hope. So therefore we didn't really realise he was going to die as quickly as he did. (GP 11)

There was a shared feeling among interviewees, that specialists should provide clear messages about patients' prognoses, so that GPs can adopt an appropriate management plan. At present, hospital specialists were perceived as failing to take responsibility for identifying patients as end of life, and this had a detrimental impact on primary care.

I feel that it should be made compulsory for the secondary care, tertiary care sectors, when they discharge, or when they're seen in the patient clinic, [to] prognosticate, … then we can initiate also, the discussion with the patient, in a much more positive way. (GP 12)

## Collaborative care pathways

Supporting patients with liver disease was seen as a collaborative effort, with GPs acknowledging their need for specialist guidance, particularly when managing end-of-life complications. A small number of respondents mentioned hepatic encephalopathy as a challenge in the management of end-of-life liver patients and a potential source of distress for relatives. The interviewees suggested that they would benefit from further training to deal with this complication. Ascites was the most commonly mentioned symptom experienced by patients with end-stage liver disease, requiring drainage in hospital. Experiences of GPs in this study suggest that ease of access to this procedure was highly variable. In some areas, pathways had been negotiated and patients could be directly admitted to an appropriate ward. In others, GPs described their concern at having to send patients to accident and emergency departments. Failure to arrange prompt access to treatment caused distress and was a major source of adverse experiences during end-of-life care.

> We had a nightmare. He was building up litres of ascitic fluid on his tummy every week or week to 10 days, and every time the hospital had to send him acutely, new, to A&E and he had to sit in A&E for hours. I was speaking to the liver specialist. He needed regular reviews and eventually they agreed to do it 2 weekly but even that wasn't enough, it was building up and he was ending up going in as an emergency every week. (GP 11)

Where appropriate care pathways were not in place, interviewees suggested that they were needed, to reassure the patient and GPs that support is available when required.

A number of participants suggested that a specialist nurse may hold the key to more collaborative management of liver patients. They could act as an intermediary between primary and secondary care, negotiating priorities and ensuring effective and easy communication.

> It often helps when there is direct access to, say, a nurse specialist in a field, or there is some other point of contact in secondary care that say a family or the patient themselves can call directly for advice. (GP 14)

Although some GPs had encountered specialist nurses working in this type of role, this was not a common experience. Unfavourable comparisons were made between the services available for patients with liver disease and other conditions, such as cancer. Participants highlighted the potential benefit to patients and families, of having a specialist point of contact in the community, including prompt access to advice and alleviation of fears and concerns.

## Social relationships and consequences

GPs in this study argued that people with liver disease had many of the same primary care needs as patients with other life-limiting conditions. However, the severity of symptoms in end-of-life liver disease was felt to be different. Some of the GPs acknowledged the potentially damaging impact on the patient's family, of seeing their relative die at home.

> I think there is quite a strong push to keep people at home. Whether that's right or wrong, I don't know really. If they've got ascites or portal hypertension, you know, they've got the risk of vomiting blood and all the rest of it. Or they have been vomiting blood. I'm not massively keen on keeping people at home because it's just a rubbish picture in the mind of everybody, I think, you know, the family left behind. (GP 17)

Families were perceived to be in need of support themselves, which was an additional role for primary care. GPs described examples of relatives requiring frequent contact and reassurance as the patient's condition deteriorated. The GPs in this study differed in their attitudes towards these demands. Some took a holistic view to the management of palliative patients, believing that these were part of the standard practice of primary care.

> I think when we talk about palliative care it's not just a single person who's the patient, it's about supporting and managing the family and helping them through that bereavement stage because it starts right at the diagnosis and they have to go through that journey. Death is a part of life and giving them that support. (GP 10)

However, others felt that attending to the needs of patients' families was an extra burden on their already overstretched resources.

Limited social support and unfavourable social circumstances were often mentioned as significant issues for patients with liver disease, particularly when alcohol or drug misuse were factors. Several GPs referred to the 'chaotic' lifestyles of this patient group and resulting vulnerability to social isolation. Behaviours associated with addiction were perceived to lead to the breakdown of the patients' social networks, leaving few, if any people to provide support or care.

> The demographics of the alcohol dependent ones, who have often, for various reasons and due to the nature of their disease, have become quite isolated, they have not got many people around them and so they don't have that support. They require much more organisation and support in the background, so we make sure that they do have that support. (GP 10)

Without alternative sources of support, socially isolated patients were believed to place extra demands on GPs and other health services. Even when social networks were maintained, there could be a dearth of responsible caregivers, as friends and family often shared the problems of addiction and poor health.

I can think of a couple of our households where maybe spouses and partners may have liver cirrhosis themselves. I can think of two couples—well, one person who died 2 years ago. His wife has chronic liver disease as well. (GP 4)

One of the most important consequences of social isolation was that patients had fewer choices over where they spent the end of their life. Without anyone to monitor their condition, they were more likely to be admitted to hospital and die there.

Liver disease is a potentially stigmatising condition, particularly when the underlying cause is alcohol or substance misuse. Several of the GPs suggested that there is often an assumption within the patient's community that liver disease is self-induced and they were culpable, even when substance misuse or alcohol are not factors.

I think it's a huge problem for people that have liver disease and look like they have liver disease and people assume it's related to alcohol when, in fact, it might be due to auto-immune causes or other forms of cancer or something like that or hepatitis as well. (GP 23)

This assumed culpability has implications for the degree of support and sympathy that the patient and their families receive. GPs also suggested that stigma could hinder patients' acceptance of their prognosis, which in turn made the management of their condition more challenging. As such, care of liver patients should include psychological and social services.

I think, inevitably and sadly, there is a stigma associated with liver disease, and hence, that's why the psychological support is really important. (GP 25)

However, some commented that stigmatisation occurred early in the patients' illnesses, and to address this, changes would be needed well before end-of-life care was being considered.

## DISCUSSION

This study provides insights into the challenges faced by general practitioners providing end-of-life care for patients with chronic liver disease. Many GPs expressed a desire to be more closely involved but identified a number of factors that constrained their ability to contribute. These fell into three main areas: those relating directly to the condition, (symptom management and the need to combine a palliative care approach with ongoing medical interventions); issues arising from patients' social circumstances (stigma, social isolation and the social consequences of liver disease) and deficiencies in the organisation and delivery of services. Collaborative working with support from specialist hospital clinicians was regarded as essential, with GPs acknowledging their own lack of experience and expertise in this area.

A majority of interviewees had little direct experience of patients dying of liver disease and as a consequence, they may not have been familiar with all the management challenges of end-stage liver disease. For example, hepatic encephalopathy is a common concern in the care of end-stage liver patients, yet few of the interviewees discussed it. This is not surprising, as primary care clinicians would seldom have responsibility for managing hepatic encephalopathy if they are not dealing day to day with end-of-life care for liver disease patients.

### Strengths and limitations
To our knowledge, this is the first study to focus on the experiences of primary care physicians in managing patients with end-stage liver disease. Our interviewees were drawn from rural and urban areas in five different regions in England and working with a diverse range of communities. The relatively large number of GP participants and varying levels of experience, expertise and interest in the subject is a particular strength of the study. With our qualitative design, we were not seeking generalisability, but the diversity of the participants increases our confidence that we have not overlooked important issues.

The majority of interviews were conducted by telephone, which may explain the ease and speed with which we recruited participants, despite not offering any financial incentives. Use of the telephone is thought to have promoted unguarded responses, but we acknowledge that it can be more difficult to develop rapport in the absence of non-verbal cues and other facets of face-to-face communication. However, we do not believe that this was a limitation, as GPs provided rich and insightful accounts of their experiences.

### Comparison with other work
Our findings are consistent with recent research from Scotland that included interviews with eight GPs along with other healthcare professionals. Communication with secondary care, lack of expertise and limited confidence in prognostication were all identified as concerns.[15] Accurate assessment of prognosis in liver disease is difficult given the unpredictable disease course. In some aspects, this is similar to other diseases characterised by episodes of decompensation, such as heart failure. However, liver disease presents the additional challenge that recompensation and improved liver function may be achieved in certain patients, such as those who achieve abstinence from alcohol. A recent review of palliative care guidelines in heart failure and chronic obstructive pulmonary disease described wide variation in how patients are identified for palliative care, and attributed this, in part, to the unpredictable disease course and the consequences for care planning.[22] In common with liver disease, acknowledgement and development of end-of-life care has been relatively recent for these conditions.[22]

The GPs in our study agreed that, at the end of life, patients with liver disease ideally need primary care and hospital specialists to work closely together. GPs are more

likely to have an established relationship with the patient and a greater understanding of their social situation and needs, whereas specialists offer expert knowledge on liver disease and treatment options. They highlighted the importance that primary care physicians place on being able to provide a coordinating role but only when supported by members of the specialist teams. Managing complex and unusual symptoms, or judging when to introduce a palliative care approach, for example, all benefit from collaboration. The advantages of a multidisciplinary approach have already been well documented in the palliative care literature.[23 24] Several recent reviews on end-stage liver disease have also advocated this approach.[4 5 14]

This study highlights the complexity of caring for patients with end-stage liver disease. Expertise in acute medicine and palliative care are essential, but patients and families also need sensitive and practical responses to their psycho-emotional and social concerns, including stigma related to the perceived self-inflictedness of the disease, social isolation and lack of income. Such generalist expertise and a holistic, person-centred approach are the foundations of primary care. Community-based services already play an important role at the end-of-life for patients with many different, complex conditions. However, this seldom includes people dying with liver disease. Greater involvement of community services would be expected to enhance the quality and appropriateness of palliative and terminal care for these patients. As the number of deaths from chronic liver disease increases, it may be increasingly necessary to limit the burden on hospital teams. Innovations, such as the development of clear patient pathways, specialist heptology nurses in the community or district nurses trained to deal with liver disease complications, all require resources. Specialist treatments such as paracentesis could be delivered in locations such as community hospitals or hospices, where they are available, to reduce disruption to patients' lives. See Box 1 for a summary of GPs' perceptions of areas for development.

In recent years, UK health policy has increasingly promoted patient choice, an ability to deliver end-of-life care in the patient's preferred location and facilitate choice in place of death are used as markers of care quality, with death at home often an implicit goal of palliative care.[25 26] GPs in this study expressed some scepticism that home death is always the best option for patients with liver disease or their families. Concerns centred around the nature of the symptoms and clinical input needed to manage them, which were potentially distressing for families to observe. Balancing the wishes of patients, families and clinical carers is a fundamental part of end of life care. More in-depth enquiry to elicit patient, family and professional views and experiences of place of death in liver disease would help to clarify the resources required to ensure death at home is acceptable and achievable.

## CONCLUSION

Our study suggests that end-of-life care for patients with liver disease requires attention. Liver disease appears to pose management challenges in end-of-life care with a combination of complicated social situations and symptoms. Services tailored for these patients should build on the similarities with other conditions but also reflect the differences. The adverse social consequences of illness for these patients and their families may be particularly significant. Further research is needed to fully understand the burden on families and services. As health services seek greater integration with social care, improving care for patients with end-stage liver disease should be a priority.

**Acknowledgements** We are grateful to the participants who generously gave their time to this study.

**Contributors** BH, EK, MH, JO and CE designed the study. HS carried out the interviews. HS undertook the main analysis supported by BH and HJ. HS, HJ and BH drafted the manuscript, and all authors commented and approved the final version.

**Funding** This research is funded by the National Institute for Health Research School for Primary Care Research (NIHR SPCR).

**Disclaimer** The views are those of the authors and not necessarily those of the NIHR, the NHS or the Department of Health.

**Competing interests** None declared.

**Patient consent** No patients were involved in this study.

**Ethics approval** Health Research Authority and Newcastle University Research Ethics (Ref 188275)

**Provenance and peer review** Not commissioned; externally peer reviewed.

**Data sharing statement** We do not have any additional unpublished data that can be shared.

| **Box 1** | **Next steps in primary end-of-life care for liver disease: general practitioner perceptions of areas for development** |
| --- | --- |

► Education and training in symptom management for end-stage liver disease.
► Collaborative care pathways between primary care and hepatologists.
► Service delivery that takes into account patient and family social circumstances and stigma.
► Provision of support for family caregivers.
► Consideration/research into the role of home death.

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
