## [Reviewer comments · BMJ Open]

ARTICLE DETAILS

TITLE (PROVISIONAL)	How can primary care enhance end-of-life care for liver disease? A qualitative study of general practitioners' perceptions and experiences
AUTHORS	Standing, Holly; Jarvis, Helen; Orr, James; Exley, Catherine; Husdon, Mark; Kaner, Eileen; Hanratty, Barbara

VERSION 1 - REVIEW

REVIEWER	Dr Benjamin Hudson University of Bristol University Hospitals Bristol NHS Trust UK
REVIEW RETURNED	24-Apr-2017

GENERAL COMMENTS	This is an interesting and well written paper on an important, clinically relevant and increasingly topical subject and I would definitely support its publication in a revised form. I think it will be of significant interest to both primary care and hepatology communities. Nonetheless there are a few issues that need to be addressed prior to publication. 1. Most crucially, there is no mention of hepatic encephalopathy throughout the paper. Clinically this is a key issue in the provision of palliative care in end stage liver disease. It is almost ubiquitous in some form, and it invariably impacts on the capacity of patients to be involved in end of life discussions and advance care planning in the terminal phases of illness. Palliative and supportive care measures need to run in parallel with curative care due to the uncertainties in the trajectory of decompensated cirrhosis, and the potential for development of encephalopathy. This theme is commonly highlighted in the literature - including some of the papers referenced - and has been highlighted as the most significant source of caregiver burden. It is undeniably a crucial issue in this patient cohort. Was this not commented upon in any of the interviews or within the topic guide? If not - why not? Does it relate to under-diagnosis from GPs or inexperience in recognising the syndrome? Are patients at this stage of disease exclusively managed by secondary care? Is encephalopathy not recognised by GPs or just not seen in clinical practice? A discussion of palliative care provision in end stage liver disease should include this extremely common and distressing feature - and this is the only major revision that is required - the remainder are minor revisions. 2. The introduction states that transplantation is the only curative treatment in end-stage liver disease. This statement is somewhat dated - for example - antiviral therapy for hepatitis C and abstinence from alcohol can cause recompensation from even very advanced
---

	disease It is an important point, as the decision that a patient is unsuitable for transplantation does not equate to inevitable terminal decline - making the decisions around instituting palliative care difficult - and the need for parallel palliative/curative approaches pertinent. 3. "In addition, GP care for patients dying with liver disease is not rated highly by bereaved relatives" - I am not sure if this is correct. My recollection of the VOICES data was that it did not specifically relate to GP care but overall care and co-ordination of care. Can this be checked. 4. I accept the limitations on word count - but a bit more detail to convince me that this is truly purposive as opposed to opportunistic sampling would be helpful. How many GPs were "screened" in the first round vs the number that went on to interview? How did you determine areas of high and low prevalence for your sampling? What features of consenting GPs were specifically targeted in selecting your second round sample? 5. "A thematic analysis was conducted based on the approach of Glaser and Strauss". You quote methodology for grounded theory, however describe a thematic analysis (granted some of the principles of grounded theory included). Nonetheless, I am not sure the reference you have provided describes your methodology accurately. 6. "Patients with a primary liver condition are often younger than the typical palliative care patient. It is likely that clinicians are more reluctant to give up on treatment for younger patients". I think both of these statements need a reference in the literature. I would also persist with the point that palliative care does not equate to giving up on curative treatment - if this is the perception that has arisen from the data this is an important point - and needs to come across more clearly. 7. There are really relevant issues raised about i) place of death in liver disease - and that death at home is not necessarily a surrogate for quality of care ii) The difficulty of stigma in liver disease to providing and accessing care. There is a burgeoning wider literature on these topics - and it would be supportive of the argument if these could be referred to. I think more attention could be paid to the POD issue in the abstract as it is very relevant to how we design services. I look forward to reading the revision.
--	---

REVIEWER	Manisha Verma, MD, MPH Director, Research Department of Transplantation/ Hepatology Einstein Healthcare Network Philadelphia, PA 19141 USA
REVIEW RETURNED	18-May-2017

GENERAL COMMENTS	This is a great article and provides insight into an important issue. Palliative care is much needed for patients with Liver disease, and currently there are no standard guidelines and resources to bring palliative care for this specific population.
---

	Your summation of themes brings together 3 key focus points to be embraced for liver disease population care. I suggest adding a few sentences under a sub-heading “ Future Steps”, and delineate what key things need to happen to overcome the hurdles faced by primary care providers. This section will make your article very useful; to not only know the deficits but also some proposals to improve the deficiency. An integrated approach in a collaborative coordinated manner will overcome most of the obstacles. It would be interesting to add- whether rural providers faced different challenges than urban providers, given the deficiency in palliative care workforce. A random picking of primary care providers from a list of providers in given regions of target can provide a generalizable result. Very few or none of the interviewed providers had been involved in direct care of patients dying due to liver disease; this is a major limitation to generalizability and must be added. Most of the patients with ESLD die in hospitals, as every attempt is usually taken to provide a curative treatment, i.e. Liver Transplant, but it is not available for all. The governing body decides and prioritizes who gets a transplant in most countries including US and UK. You could add this to describe limited prognostication for liver disease. There is not much emphasis on limited knowledge or limited expertise in palliative care, which could likely be overcome by adding some educational resources for primary care clinicians to learn more about palliative care in general. You could add this to your future plan section. Development of care pathways must be emphasized more, as this is the way which can help establish standardized guidelines. Overall, you have discussed a very important issue which needs urgent attention. Thank you.
--	---

VERSION 1 – AUTHOR RESPONSE

Reviewer 1

1. ... there is no mention of hepatic encephalopathy throughout the paper. Clinically this is a key issue in the provision of palliative care in end stage liver disease. It is almost ubiquitous in some form, and it invariably impacts on the capacity of patients to be involved in end of life discussions and advance care planning in the terminal phases of illness. Palliative and supportive care measures need to run in parallel with curative care due to the uncertainties in the trajectory of decompensated cirrhosis, and the potential for development of encephalopathy. This theme is commonly highlighted in the literature - including some of the papers referenced - and has been highlighted as the most significant source of caregiver burden. It is undeniably a crucial issue in this patient cohort. Was this not commented upon in any of the interviews or within the topic guide? If not - why not? Does it relate to under-diagnosis from GPs or inexperience in recognising the syndrome? Are patients at this stage of disease exclusively managed by secondary care? Is encephalopathy not recognised by GPs or just not seen in clinical practice? A discussion of palliative care provision in end stage liver disease should include this extremely common and distressing feature - and this is the only major revision that is required - the remainder are minor revisions.

Thanks to Reviewer 1 for highlighting our omission. The topic guide asked interviewees generally about challenges and complications in end-stage liver disease, but it did not refer to specific symptoms or complications. This approach was chosen to draw out the interviewees existing level of knowledge. However, we have now reviewed our data to explore any mentions of hepatic encephalopathy. This occurred in a very small proportion of the interviews, and has been included in the findings section. In addition, we have added some discussion of the implications of this finding. We interpret the limited reference to hepatic encephalopathy as an indication that these clinicians are

currently not immersed in end of life care for this patient group.

2. The introduction states that transplantation is the only curative treatment in end-stage liver disease. This statement is somewhat dated - for example - antiviral therapy for hepatitis C and abstinence from alcohol can cause recompensation from even very advanced disease. It is an important point, as the decision that a patient is unsuitable for transplantation does not equate to inevitable terminal decline - making the decisions around instituting palliative care difficult - and the need for parallel palliative/curative approaches pertinent.

We apologise for this misleading sentence as we did not intend to suggest that terminal decline is inevitable for people who do not receive transplants. We have amended the wording in the introduction as follows:

'Transplantation is an option only for selected patients,⁹ with some evidence suggesting that patients who are considered and rejected for organ transplantation, are unlikely to receive any palliative care.¹⁰'

3. In addition, GP care for patients dying with liver disease is not rated highly by bereaved relatives" - I am not sure if this is correct. My recollection of the VOICES data was that it did not specifically relate to GP care but overall care and co-ordination of care. Can this be checked.

There is a section in VOICES that refers to GP care and we have inserted a reference to a report that describes the VOICES data on GP care for people living with liver disease:

Kendrick E. Getting it Right Improving End of Life Care for People Living with Liver Disease. London: National End of Life Care Programme, 2013

4. I accept the limitations on word count - but a bit more detail to convince me that this is truly purposive as opposed to opportunistic sampling would be helpful. How many GPs were "screened" in the first round vs the number that went on to interview? How did you determine areas of high and low prevalence for your sampling? What features of consenting GPs were specifically targeted in selecting your second round sample?

We agree with Reviewer 1 that our description of the sampling was vague. We have amended the text in the methods section to describe the sampling procedure in more detail as follows:

'Following the first phase of interviews, participants were purposively sampled in order to provide a wide range of clinical experience and degree of familiarity with liver disease. To do this, we expanded the study to include two additional geographical sites and worked with co-ordinators at the Clinical Research Networks to target practices in a variety of rural and urban locations, as well as areas of high and low prevalence of liver disease and substance misuse.'

5. A thematic analysis was conducted based on the approach of Glaser and Strauss". You quote methodology for grounded theory, however describe a thematic analysis (granted some of the principles of grounded theory included). Nonetheless, I am not sure the reference you have provided describes your methodology accurately.

Reviewer 1 is correct, we employed the principles of grounded theory in an applied thematic analysis. We have provided more appropriate references for our methodology:

Braun V, Clarke, V. Using thematic analysis in psychology. *Qualitative Research in Psychology* 2006;3(2):77-101. doi: 10.1191/1478088706qp063oa
Ritchie J, Lewis J. *Qualitative Research Practice: A Guide for Social Science Students and Researchers* London: SAGE Publications; 2003

6. "Patients with a primary liver condition are often younger than the typical palliative care patient. It is likely that clinicians are more reluctant to give up on treatment for younger patients". I think both of these statements need a reference in the literature. I would also persist with the point that palliative care does not equate to giving up on curative treatment - if this is the perception that has arisen from the data this is an important point - and needs to come across more clearly.'

Reviewer 1 makes an important point and we have included references to support our statements:

- Office for National Statistics. Deaths registered in England and Wales (Series DR): 2015. <https://www.ons.gov.uk/peoplepopulationandcommunity/birthsdeathsandmarriages/deaths/bulletins/deathsregisteredinenglandandwalesseriesdr/2015>, 2016.
- Department of Health. The impact of patient age on clinical decision-making in oncology. In: Department of Health, ed. <https://www.gov.uk/government/publications/the-impact-of-patient-age-on-clinical-decision-making-in-oncology>, 2012.
- Thompson GN, Chochinov, H.M., Wilson, K. G., McPherson, C. J., Chary, S., O'Shea, F. M., Kuhl, D. R., Fainsinger, R. L., Gagnon, P.R., Macmillan, K. A. Prognostic Acceptance and the Well-Being of Patients Receiving Palliative Care for Cancer. *Journal of Clinical Oncology* 2009;27(34):5757-62.

We have also reviewed our text to ensure that we have not mistakenly given the impression of a dichotomy between palliative and curative care.

7. There are really relevant issues raised about i) place of death in liver disease - and that death at home is not necessarily a surrogate for quality of care ii) The difficulty of stigma in liver disease to providing and accessing care. There is a burgeoning wider literature on these topics - and it would be supportive of the argument if these could be referred to. I think more attention could be paid to the POD issue in the abstract as it is very relevant to how we design services.

We agree that place of death is a central issue to end of life care for all patients and choice in this matter is reliant on the support available in community settings. However, we did not set out to conduct a study around place of death – which is an important issue in its own right. And we have no data from patients or relatives, which would be essential in a robust study of place of death preferences in liver disease.

Taking care not to stray too far beyond our data, we have added a short paragraph to the discussion to highlight this issue. We have focussed on the potential conflict between policies that promote home as the preferred place of care and death, and the reality for families, patients and GPs.

Reviewer: 2

1. I suggest adding a few sentences under a sub-heading "Future Steps", and delineate what key things need to happen to overcome the hurdles faced by primary care providers. This section will make your article very useful; to not only know the deficits but also some proposals to improve the

deficiency. An integrated approach in a collaborative coordinated manner will overcome most of the obstacles.

Reviewer 2 makes a very practical and useful suggestion. We are cautious about making recommendations for service redesign from a qualitative study involving only one constituency. Hence, we have added Box 1 – entitled 'Next steps in primary end of life care for liver disease: GP perceptions of areas for development'. This lists the main areas for future attention, and is referred to in the text in the discussion.

2. It would be interesting to add- whether rural providers faced different challenges than urban providers, given the deficiency in palliative care workforce. A random picking of primary care providers from a list of providers in given regions of target can provide a generalizable result.

Unfortunately, we did not have the right study design or sample size to robustly address this question. But we agree that it is an important one. Our sampling strategy was influenced by a concern that GPs in areas close to national centres of excellence in hepatology (e.g Newcastle) may have different relationships with and expectations of hepatologists.

3. Very few or none of the interviewed providers had been involved in direct care of patients dying due to liver disease; this is a major limitation to generalizability and must be added. Most of the patients with ESLD die in hospitals, as every attempt is usually taken to provide a curative treatment, i.e. Liver Transplant, but it is not available for all. The governing body decides and prioritizes who gets a transplant in most countries including US and UK. You could add this to describe limited prognostication for liver disease.

We feel that the lack of experience of end of life care for liver disease is a finding of our study, rather than a limitation. We have no reason to think that if we had done a large survey, we would have found areas where a majority of GPs are closely involved in this area of care. We have added a reference to criteria for access to liver transplantation, and linked to Reviewer 1's comments about the role of transplantation.

4. There is not much emphasis on limited knowledge or limited expertise in palliative care, which could likely be overcome by adding some educational resources for primary care clinicians to learn more about palliative care in general. You could add this to your future plan section.

Thank you – this is one of our most important points – it is highlighted in Box 1.

5. Development of care pathways must be emphasized more, as this is the way which can help establish standardized guidelines.

We agree this is likely to be the most appropriate way forward, and have mentioned this throughout, where it has appeared in our data.

VERSION 2 – REVIEW

REVIEWER	Ben Hudson University Hospitals Bristol NHS Foundation Trust University of Bristol, School of Clinical Sciences
REVIEW RETURNED	28-Jun-2017

GENERAL COMMENTS	Thank you to the authors for providing a thorough and thoughtful response to the earlier reviews - I think the manuscript has been improved by this revision and I enjoyed reading it. In particular, I think the added paragraph in the discussion regarding the scarcity of data on HE provides a sensible interpretation. I would like to see this published - it covers an important topic and is well written. I would suggest 2 further minor revisions prior to publication. 1. I remain unconvinced by that the sampling methodology described (which was updated in the revision) achieves its stated aim. Please see quoted section below. I don't think the second sentence backs up the first. Targeting areas of high prevalence (how was this determined?) geographically, and based on whether a practice was rural or urban, does not equate to a purposive sample based on an individual's clinical experience of managing liver disease. Also typo on geographically. Was your aim actually to get a wide geographical representation? If so there are specific methodologies for this. It still seems like more of an opportunistic sample to me - but I suppose this is a subjective opinion (and I certainly wouldn't reject on this basis). "Following the first phase of interviews, participants were purposively sampled in order to provide a wide range of clinical experience and degree of familiarity with liver disease. To do this, we expanded the study to include two additional geographical sites and worked with co-ordinators at the Clinical Research Networks to target practices in a variety of rural and urban locations, as well as areas of high and low prevalence of liver disease and substance misuse 2. Minor point, but the sentence - "compared to cancer patients, people with liver disease are less likely to discuss and plan their end of life care" - stretches the interpretation of the evidence quoted somewhat. The reference given discusses studies around DNAR decisions in ESLD, and comments that data around the issue of ACP are scarce (it does say that ACP issues SEEM to be discussed less frequently). Whilst the sentiment is true - I would prefer this sentence to be qualified slightly (again subjective/pedantic - apologies). Overall however, this is a thought provoking piece of work, which I am sure will make a valuable addition to the literature.
--

REVIEWER	Manisha Verma Einstein Medical Center, Philadelphia PA USA
REVIEW RETURNED	18-Jun-2017

GENERAL COMMENTS	All the queries have been addressed satisfactorily.
---

VERSION 2 – AUTHOR RESPONSE

Reviewer 1

1. I remain unconvinced by that the sampling methodology described (which was updated in the revision) achieves its stated aim. Please see quoted section below. I don't think the second sentence backs up the first. Targeting areas of high prevalence (how was this determined?) geographically, and based on whether a practice was rural or urban, does not equate to a purposive sample based on an individual's clinical experience of managing liver disease. Also typo on geographically. Was your aim actually to get a wide geographical representation? If so there are specific methodologies for this. It still seems like more of an opportunistic sample to me - but I suppose this is a subjective opinion (and I certainly wouldn't reject on this basis).

We thank you for noticing the typographical error in this section, this has now been corrected. We have also amended the section on sampling to provide greater clarity on our intention to use heterogeneous purposive sampling to account for variation in practice size and locality as well as doctors' experience in managing end of life liver disease. We hope this section is now clear.

'A heterogeneous purposive sampling approach was employed to ensure that a variety of perspectives and experiences of management of liver disease were sampled e.g. previous management of an end of life liver patient, views on whether management should be primary care or secondary care led, as well as a range of practice size and locality. Participants were recruited via National Institute for Health Research Clinical Research Networks (CRN) and local networks of GP practices in London, Thames Valley, Wessex, Yorkshire and the North East of England. Sampling began with one CRN and was expanded during the course of the study to include four additional areas. Co-ordinators at the CRNs were utilised to target practices in a variety of rural and urban locations, as well as areas of high and low prevalence of liver disease and substance misuse. Email invitations were sent to GP practices within these networks, and GPs who wished to participate then contacted the research team.'

2. Minor point, but the sentence - "compared to cancer patients, people with liver disease are less likely to discuss and plan their end of life care" - stretches the interpretation of the evidence quoted somewhat. The reference given discusses studies around DNAR decisions in ESLD, and comments that data around the issue of ACP are scarce (it does say that ACP issues SEEM to be discussed less frequently). Whilst the sentiment is true - I would prefer this sentence to be qualified slightly (again subjective/pedantic - apologies).

We thank you for this comment. We did not mean to mislead in our presentation of these data, and we have rephrased the sentence for greater clarity.

"Research suggest that people with liver disease are less likely to be involved in end of life discussions and planning than cancer patients, though data are limited. 11" Reviewer 1

1. I remain unconvinced by that the sampling methodology described (which was updated in the revision) achieves its stated aim. Please see quoted section below. I don't think the second sentence backs up the first. Targeting areas of high prevalence (how was this determined?) geographically, and based on whether a practice was rural or urban, does not equate to a purposive sample based on an individual's clinical experience of managing liver disease. Also typo on geographically. Was your aim actually to get a wide geographical representation? If so there are specific methodologies for this. It still seems like more of an opportunistic sample to me - but I suppose this is a subjective opinion (and I certainly wouldn't reject on this basis).

We thank you for noticing the typographical error in this section, this has now been corrected. We have also amended the section on sampling to provide greater clarity on our intention to use heterogeneous purposive sampling to account for variation in practice size and locality as well as doctors' experience in managing end of life liver disease. We hope this section is now clear.

'A heterogeneous purposive sampling approach was employed to ensure that a variety of perspectives and experiences of management of liver disease were sampled e.g. previous management of an end of life liver patient, views on whether management should be primary care or secondary care led, as well as a range of practice size and locality. Participants were recruited via National Institute for Health Research Clinical Research Networks (CRN) and local networks of GP practices in London, Thames Valley, Wessex, Yorkshire and the North East of England. Sampling began with one CRN and was expanded during the course of the study to include four additional areas. Co-ordinators at the CRNs were utilised to target practices in a variety of rural and urban locations, as well as areas of high and low prevalence of liver disease and substance misuse. Email invitations were sent to GP practices within these networks, and GPs who wished to participate then contacted the research team.'

2. Minor point, but the sentence - "compared to cancer patients, people with liver disease are less likely to discuss and plan their end of life care" - stretches the interpretation of the evidence quoted somewhat. The reference given discusses studies around DNAR decisions in ESLD, and comments that data around the issue of ACP are scarce (it does say that ACP issues SEEM to be discussed less frequently). Whilst the sentiment is true - I would prefer this sentence to be qualified slightly (again subjective/pedantic - apologies).

We thank you for this comment. We did not mean to mislead in our presentation of these data, and we have rephrased the sentence for greater clarity.

"Research suggest that people with liver disease are less likely to be involved in end of life discussions and planning than cancer patients, though data are limited. 11"

VERSION 3 – REVIEW

REVIEWER	Ben Hudson University Hospitals Bristol NHS Foundation Trust University of Bristol
REVIEW RETURNED	13-Jul-2017

GENERAL COMMENTS	Thank you - my previous comments have been addressed satisfactorily in this revision. Excellent article - well done.
--